# Unraveling Motion Uncertainty for Local Motion Deblurring

Zeyu Xiao*
University of Science and Technology
of China & National University of
Singapore
Hefei, China & Singapore
zeyuxiao@mail.ustc.edu.cn

Zhihe Lu†
National University of Singapore
Singapore
zhihelu@nus.edu.sg

Michael Bi Mi
Huawei International Pte Ltd
Singapore

Zhiwei Xiong
University of Science and Technology
of China
Hefei, China
zwxiong@ustc.edu.cn

Xinchao Wang‡
National University of Singapore
Singapore
xinchao@nus.edu.sg

## Abstract

In real-world photography, local motion blur often arises from the interplay between moving objects and stationary backgrounds during exposure. Existing deblurring methods face challenges in addressing local motion deblurring due to (i) the presence of arbitrary localized blurs and uncertain blur extents; (ii) the limited ability to accurately identify specific blurs resulting from ambiguous motion boundaries. These limitations often lead to suboptimal solutions when estimating blur maps and generating final deblurred images. To that end, we propose a novel method named Motion-Uncertainty-Guided Network (MUGNet), which harnesses a probabilistic representational model to explicitly address the intricacies stemming from motion uncertainties. Specifically, MUGNet consists of two key components, *i.e.*, motion-uncertainty quantification (MUQ) module and motion-masked separable attention (M2SA) module, serving for complementary purposes. Concretely, MUQ aims to learn a conditional distribution for accurate and reliable blur map estimation, while the M2SA module is to enhance the representation of regions influenced by local motion blur and static background, which is achieved by promoting the establishment of extensive global interactions. We demonstrate the superiority of our MUGNet with extensive experiments. The code is publicly available at: https://github.com/zeyuxiao1997/MUGNet.

## CCS Concepts

• **Computing methodologies → Reconstruction**.

---

*This work is done when Zeyu is a visiting student at National University of Singapore.
†Zeyu and Zhihe contribute equally.
‡Corresponding author.

---

## Keywords

Image restoration, Image deblurring, Local deblurring, Motion uncertainty

**ACM Reference Format:**
Zeyu Xiao, Zhihe Lu, Michael Bi Mi, Zhiwei Xiong, and Xinchao Wang. 2024. Unraveling Motion Uncertainty for Local Motion Deblurring. In *Proceedings of the 32nd ACM International Conference on Multimedia (MM '24), October 28-November 1, 2024, Melbourne, VIC, Australia.* ACM, New York, NY, USA, 10 pages. https://doi.org/10.1145/3664647.3681239

## 1 Introduction

Image deblurring is a critical task in the fields of image processing, aiming to recover clear and detailed images from blurry ones caused by various factors, such as camera shake [29, 34], object motion [30, 43, 56], and being out-of-focus [1, 2, 20]. Motion blur removal presents a significant challenge in image deblurring, primarily due to arbitrary localized blurs and uncertain blur extents, sparking considerable interest in the research community. This problem can be broadly categorized into two main types: global motion deblurring and local motion deblurring. Global motion blur uniformly affects the entire image, while local motion blur is constrained to specific regions within the image, appearing selectively in certain parts of the scene. Advances in image deblurring techniques have the potential to benefit numerous fields, including photography, multimedia, medical imaging, remote sensing, surveillance, and computer vision, by enabling clearer image representations and facilitating more accurate image-based tasks and analyses.

A straightforward strategy is to apply global deblurring techniques to local blurred images to achieve effective local motion deblurring. However, this approach may not yield optimal results (see Figure 1), as local motion blur often presents distinct characteristics compared to global motion blur. Specifically, local motion blur is characterized by abrupt changes at object boundaries, while the background remains clear. This discrepancy poses challenges for global deblurring methods and may result in undesired artifacts in originally sharp regions. Therefore, specialized methods that explicitly address the unique problems of local motion blur are essential to ensure accurate and artifact-free deblurring in such scenarios, which has yet to gain limited attention.

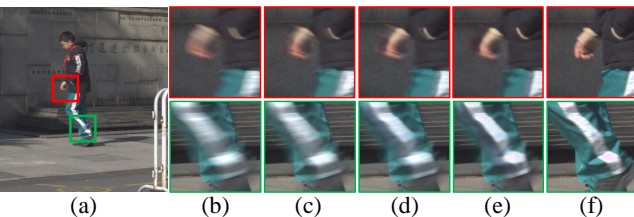

| (a) | (b) | (c) | (d) | (e) | (f) |

**Figure 1: Examples of local motion deblurring. (a) Locally blurred image, (b) blurred patches, (c) results of MIMO-UNet [6], (d) results of LBAG [21], (e) results of our proposed MUGNet, and (f) ground-truth images. MUGNet achieves superior results, showcasing its effectiveness in local motion deblurring. Please zoom in for better visualization.**

Thanks to [21], the first paired real, local motion blur dataset – ReLoBlur, and an end-to-end framework called LBAG for local motion deblurring have been proposed. Specifically, LBAG incorporates a gated structure to alleviate the deblurring impact on non-blurred regions, preventing unwanted distortion in the background and static objects. While LBAG represents a significant step forward in this direction, it still lacks local blur detection. This is because LBAG employs a gated structure to directly multiply estimated blur masks with blurry images, often resulting in sub-optimal results, as shown in Figure 1. There are several core challenges for identifying local motion blur: (i) addressing abrupt changes at object boundaries, where blur intensity can vary significantly within a confined spatial area; (ii) capturing the intricate and irregular patterns often exhibited by local motion blur, posing challenges for modeling and prediction; (iii) dealing with occlusions and object movements, which introduce additional complexity to accurately identifying local motion blur.

In this paper, we introduce a novel method known as the Motion-Uncertainty-Guided Network (MUGNet) to address the intricate challenges of local motion deblurring. Given the complexities posed by random blur localization, elusive extents, and indistinct motion boundaries, our MUGNet involves estimating motion uncertainty maps by applying the motion-uncertainty quantification (MUQ) module. To achieve this, we draw inspiration from the Bayesian probability theory [3, 12, 14, 15, 23, 25, 51, 59], and structure our MUQ module as a probabilistic representational model, steering clear of pixel-level blur mask estimations and instead learning probability distributions. Concretely, by drawing $K$ samples from these acquired distributions, we are able to generate initial estimates and quantify the degrees of motion uncertainties. Incorporating the MUQ module significantly enhances the capability to estimate more precise blur masks, which is crucial for local motion deblurring. Furthermore, we introduce the motion-masked separable attention (M2SA) module, which uses the Multi-Dconv Head Transposed Attention (TA) to handle both blurry objects and stationary backgrounds. Instead of performing the same operations in each head, our M2SA separates the attention heads into two groups: the masked-TA (MTA) enhances the representation of blurry areas, and the global-TA (GTA) can build global interactions for generating high-quality clear images.

Utilizing our proposed Motion-Uncertainty-Guided Network (MUGNet) with the newly introduced Motion-Uncertainty Quantification (MUQ) module and Motion-Masked Separable Attention (M2SA) module, we adopt a multi-scale architecture to generate blur-free images. This integration allows MUGNet to produce remarkably enhanced results with finer details, surpassing state-of-the-art methods. To validate the efficacy of our approach, extensive experiments are conducted on the ReLoBlur testing dataset [21], demonstrating superior performance across various local motion deblurring scenarios.

The contributions of this work are summarized as follows:

- We introduce the Bayesian learning into blur mask estimation and propose the MUQ module. By explicitly quantifying motion uncertainty, the MUQ module enables accurate blur mask estimation for local motion deblurring.
- We further present the M2SA module, which divides the attention heads into two groups and enhances the representations of the blurry areas and background static regions by separately computing their attention scores via the estimated blur masks for accurate local motion deblurring.
- The MUQ module and the M2SA module together constitute our MUGNet. Extensive experiments demonstrate that the proposed MUGNet achieves superior performance.

## 2 Related Work

**Global motion deblurring.** Image deblurring poses a challenging ill-posed problem [24, 40, 44–48], seeking to recover clear images from their blurry counterparts. A range of regularization priors, including heavy-tailed gradient, sparse kernel, $l_0$ gradient, normalized sparsity, and dark channels, have been developed to guide the solution space towards sharp latent images [9, 17, 28, 33, 49]. In recent times, the advancement of deep learning has brought transformative progress to image deblurring, notably within the domain of global deblurring [31, 36, 50, 54, 57, 58, 61]. Pioneering deep global motion deblurring efforts have embraced Convolutional Neural Networks (CNNs) as foundational components, yielding substantial improvements in image quality. Among these, DeepDeblur [27] stands out, employing a multi-scale CNN architecture with residual blocks to expedite convergence. Innovations like DeblurGAN [18] and DeblurGAN-v2 [19] have harnessed Generative Adversarial Networks (GANs) and perceptual loss to enhance subjective image quality. HINet [5]h leverages Instance Normalization for performance gains. Harnessing the potential of vision Transformers [8] to capture long-range dependencies, their application in global deblurring tasks has ignited substantial interest. For instance, Uformer [41] employs window-based self-attention coupled with a learnable multi-scale restoration modulator, effectively capturing local and global dependencies. Concurrently, Restormer [52] exploits TA and a feed-forward network, enabling intricate long-range pixel interactions. Unlike these methods, in this paper, the proposed MUGNet aims to deblur locally blurred images.

**Local motion deblurring.** Local motion deblurring, less explored, tackles blurs in specific areas caused by object movements with stationary cameras[21, 22]. A notable stride in this direction has been taken by [21], who have undertaken a noteworthy endeavor in curating the pioneering ReLoBlur dataset, a foundational resource

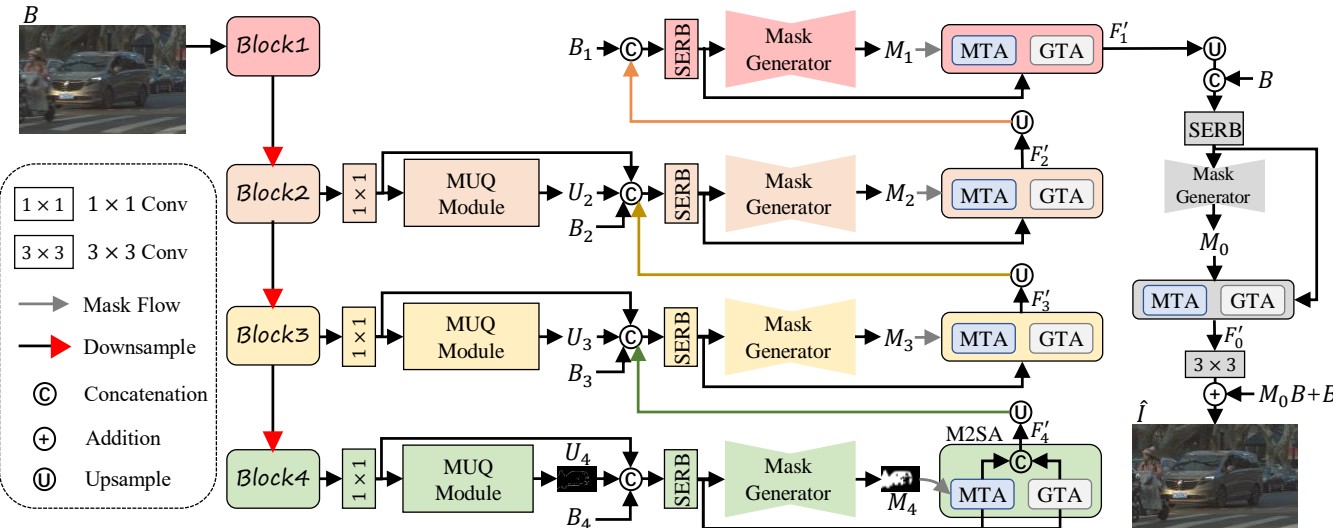

**Figure 2: An overview of the proposed MUGNet. MUGNet is built on a multi-scale encoder-decoder structure.**

pivotal for propelling the frontiers of local motion deblurring research. Within this context, they have introduced LBAG, an end-to-end framework meticulously tailored to tackle the intricacies of local motion deblurring. Very recently, Li *et al.* [22] propose LMD-ViT. LMD-ViT is a sparse vision Transformer for restoring images affected by local motion blurs. LMD-ViT is built upon the adaptive window pruning Transformer blocks, which employ blur-aware confidence predictors to estimate the level of blur confidence in the feature domain. Building upon this, we present MUGNet, a novel method specifically designed to achieve accurate blur map estimation and enhance local motion deblurring capabilities. Our method harnesses the power of the MUQ and M2SA modules to address the challenges posed by local motion deblurring.

## 3 Method

Given a blurry image $B \in \mathbb{R}^{H \times W \times 3}$, the proposed MUGNet aims to reconstruct a high-quality clear image $\hat{I} \in \mathbb{R}^{H \times W \times 3}$, which should be close to the ground truth image $I^{GT} \in \mathbb{R}^{H \times W \times 3}$. $H$ and $W$ denote the height and width.

We utilize a multi-scale encoder to extract the features of the blurry image since multi-scale structures prove effective in image deblurring. Specifically, we feed $B$ into the encoder to generate multi-scale feature maps from four stages based on ResNet, which are denoted as $\{F_i\}_{i=1}^4$. Consequently, $F_1$ is with spatial size $\frac{H}{4} \times \frac{W}{4}$ and $F_4$ is with spatial size $\frac{H}{32} \times \frac{W}{32}$. To achieve a better trade-off between efficiency and performance, we first connect a $1 \times 1$ convolution with 32 channels to the feature maps at each level and obtain $\{F_i^c\}_{i=1}^4$. At levels 2, 3, and 4, we separately input $\{F_i^c\}_{i=2}^4$ into the MUQ module to generate corresponding motion uncertainty maps $\{U_i\}_{i=2}^4$. Subsequently, the estimated $\{U_i\}_{i=2}^4$, rescaled blurry images $\{B_i\}_{i=2}^4$, and the upscaled feature are concatenated and fed to SE-Residual Blocks (SERBs), followed by mask generators to estimate blur masks $\{M_i\}_{i=2}^4$. The mask generator consists of a $3 \times 3$ convolution, followed by the sigmoid operation. Note that we

do not binarize the blur maps but keep them as continuous maps ranging from 0 to 1. Then, the M2SA module is applied at each feature level to enhance the distinction between blur regions and static background regions, generating $\{F_i^{'}\}_{i=2}^4$. At levels 1 and 0, $\{F_i^{'}\}_{i=0,1}$ and blurry inputs $B_1$ and $B$ are concatenated and fed to SERBs, followed by mask generators to generate $\{M_i\}_{i=0,1}$. The M2SA module is applied to generate $\{F_i^{'}\}_{i=0,1}$. To obtain the final clear result, $F_0^{'}$ is fed to a $3 \times 3$ convolution. It is then added in a residual manner with the product of $M_0$ and $B$, resulting in $\hat{I}$

$$\hat{I} = \text{Conv}(F_0^{'}) + M_0 B + B, \tag{1}$$

where $\text{Conv}(\cdot)$ denotes the $3 \times 3$ convolution.

### 3.1 Motion Uncertainty Quantification Module

Accurately estimating blur masks is crucial for achieving precise deblurring results. In practice, however, the task of identifying and estimating blur masks for local motion blur is notably challenging, primarily due to its distinctive and unpredictable characteristics. Therefore, simply feeding features into the following convolution layers to estimate blur masks is not optimal. Inspired by Bayesian probability theory [3, 12, 14, 15, 23, 60], we propose to estimate the motion uncertainty maps using the MUQ module and then generate more accurate blur masks. We employ uncertainty estimation to derive blur masks, aligning with local blurry artifacts' distinctive characteristics.

As shown in Figure 3, we design the MUQ module as a probabilistic representational model to measure motion uncertainty. Therefore, what the MUQ module delivers for each pixel (*e.g.*, the pixel $p$) is a distribution parameterized by mean $\boldsymbol{\mu}_p$ and variance $\boldsymbol{\sigma}_p$ instead of a scalar (*e.g.*, a score). Following [12], we model the distribution of outputs at each pixel as Gaussian, and therefore, the prediction of the MUQ module is a random variable. We expect that the blur score at the position $p$ can be drawn from the learned distribution: $u_p \sim \mathcal{N}(\boldsymbol{\mu}_p, \boldsymbol{\sigma}_p)$, where $\boldsymbol{\mu}_p$ and $\boldsymbol{\sigma}_p$ are learned by a

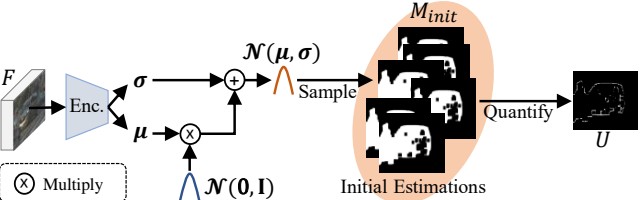

**Figure 3: Illustration of the MUQ module. The MUQ module works as a probabilistic model for motion uncertainty quantification, which is composed of a feature encoder.**

two-branch encoder consists of a $1 \times 1$ convolution

$$\boldsymbol{\mu} = \text{Enc}_{\mu}(F), \boldsymbol{\sigma} = \text{Enc}_{\sigma}(F), \tag{2}$$

where $\boldsymbol{\mu} \in \mathbb{R}^{h \times w \times 1}$ and $\boldsymbol{\delta} \in \mathbb{R}^{h \times w \times 1}$ denote the mean map and variance map, respectively. $h$ and $w$ denote the height and width of $F$, and we omit the subscript $i$. As observed in existing studies [10, 12, 13], using random samples to train the MUQ module can lead to the lack of error propagation from the output. To address this issue, taking inspiration from [15], we decompose the direct sampling operation into two components: trainable and random parts. In particular, we start by randomly drawing a sample $\epsilon_p$ from the standard Gaussian distribution $\mathcal{N}(0, I)$, i.e., $\epsilon_p \sim \mathcal{N}(0, I)$, and then compute the sample as $\boldsymbol{\mu} + \sigma\epsilon_p$. This approach allows gradients to propagate backward, enabling the optimization of the MUQ module.

We sample $K$ initial blur masks from the learned distribution to measure pixel-wise motion uncertainty, denoted as $M_{init} = \{m_1, \cdots, m_K\}$. According to Bayesian probability theory [3, 12, 14], we can treat $M_{init}$ as empirical samples from an approximate predictive distribution and measure how confident the model is in its prediction by computing the variance

$$U = \text{Norm}(\text{Var}(M_{init})), \tag{3}$$

where $U \in \mathbb{R}^{h \times w \times 1}$ means the motion uncertainty map, $\text{Norm}(\cdot)$ is the mean-max normalization operation and $\text{Var}(\cdot)$ denotes the operation of computing variance.

The estimated motion uncertainty map effectively guides the generation of blur maps within our proposed MUQNet framework, resulting in improved precision and quality.

## 3.2 Motion-Masked Separable Attention

Blurred regions exhibit diverse scales and frequently share visual characteristics with the background, presenting inherent challenges for local motion deblurring. Using estimated blur masks offers a valuable tool to explicitly differentiate and process the blurry objects and the static background. This approach not only enhances the feature representations but also contributes to a more effective deblurring process, ultimately leading to improved results. We therefore propose the M2SA module to utilize a subset of attention heads to independently compute attention scores for the predicted blurry areas and static background regions, enhancing the feature representation.

Our M2SA module is based on a modified version of self-attention to save computations, namely TA [52]. Given an input $X \in \mathbb{R}^{hw \times c}$ where $h$ and $w$ are respectively the height and width while $c$ is the

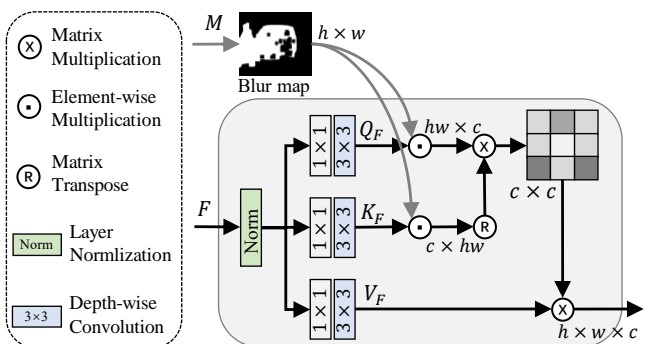

**Figure 4: Diagrammatic details of the proposed MTA in our M2SA. For simplicity, we omit the subscript $i$ indicating the $i$-th level.**

channel number, TA can be formulated as:

$$\text{TA}(Q, K, V) = V \cdot \text{Softmax}(\frac{Q^{\top}K}{\alpha}), \tag{4}$$

where $Q, K, V$ are the query, key, value matrices that can be generated by using three separate $1 \times 1$ convolutions followed by a $3 \times 3$ depthwise convolution, and $\alpha$ is a learnable scaling parameter. Eq. 4 can also be extended to a multi-head version, as done in the original self-attention [39], to enhance the feature representations.

The attention heads in the above TA are equally utilized for encoding spatial information, and we term these heads as global TA (GTA). In our M2SA, we propose introducing the estimated blur maps at each feature level into TA to better distinguish between blurry objects and static backgrounds and enhance the network's representation. To achieve this, we divide all attention heads into mask-head TA (MTA) and the regular GTA. The detail of MTA is shown in Figure 4.

To be specific, given a predicted blur map $M$, the formulation of MTA can be written as

$$MTA(Q_F, K_F, V_F) = V_F \cdot \text{Softmax}(\frac{Q_F^{\top}K_F}{\alpha_F}), \tag{5}$$

where $Q_F, K_F$ are the masked query and key matrices that can be produced by multiplying them with $M$ and $V_F$ is the value matrix without masking. In this way, the features can be refined by building pairwise relationships within blurry objects in the foreground, avoiding the influence of the background, which may contain contaminative information.

Other than the MTA heads, the second group of the heads is kept unchanged as in Eq. 4, which is used to build relationships between the foreground and background globally. The outputs of all heads are then concatenated and sent into a $3 \times 3$ convolution for feature aggregation

$$F^{'} = \text{Conv}([MTA, GTA]), \tag{6}$$

where $[\cdot, \cdot]$ is the concatenation operation.

## 3.3 Loss Functions

Following [21], we utilize several loss functions to optimize the proposed MUGNet.

**Table 1: Quantitative comparison of different methods on the ReLoBlur dataset. "PSNR$_w$", "SSIM$_w$", "PSNR$_a$", "Runtime" and "Params" denote weighted PSNR, weighted SSIM, aligned PSNR, inference time, and model parameters respectively. The best and the second best results are highlighted in bold and underline.**

| Methods | ↑PSNR | ↑SSIM | ↑PSNR$_w$ | ↑SSIM$_w$ | ↑PSNR$_a$ | Runtime | #Params | FLOPs |
|---|---|---|---|---|---|---|---|---|
| DeepDeblur [27] | 33.05 | 0.8946 | 26.51 | 0.8152 | 33.70 | 0.50s | 11.72M | 17.133T |
| DeblurGAN-v2 [19] | 33.85 | 0.9027 | 27.37 | 0.8342 | 34.30 | **0.07s** | **5.076M** | 0.989T |
| SRN-DeblurNet [36] | 34.30 | 0.9238 | 27.48 | 0.8570 | 34.88 | 0.31s | 88.67M | 8.696T |
| HINet [5] | 34.36 | 0.9151 | 27.64 | 0.8510 | 34.95 | 0.31s | 88.67M | 8.696T |
| MIMO-UNet [6] | 34.52 | 0.9250 | 27.95 | 0.8650 | 35.42 | 0.51s | 16.11M | 7.850T |
| Restormer [52] | 34.92 | 0.9265 | 29.47 | 0.8811 | - | 3.72s | 26.13M | 6.741T |
| Uformer-B [41] | 35.19 | 0.9265 | 30.22 | 0.8911 | - | 1.31s | 50.88M | 4.375T |
| LBAG [21] | 34.66 | 0.9249 | 28.25 | 0.8692 | 35.39 | 0.51s | 16.11M | 7.852T |
| LBAG+ [21] | 34.85 | 0.9257 | 28.32 | 0.8734 | 35.53 | 0.51s | 16.11M | 7.852T |
| LMD-ViT [22] | 35.42 | 0.9289 | 30.25 | **0.8938** | - | 0.56s | 54.50M | 1.485T |
| **MUGNet** | **36.31** | **0.9323** | **30.28** | 0.8898 | **36.73** | 0.66s | 24.67M | **0.627T** |

**Blur mask prediction loss $\mathcal{L}_M$.** We utilize the blur mask prediction loss to constrain our estimated blur masks effectively. Specifically, we employ the Mean Squared Error (MSE) loss to compute the loss across multiple scales, which in turn contributes to the effective regularization and refinement of our blur mask estimation. $\mathcal{L}_M$ can be denoted as

$$\mathcal{L}_M = \sum_{i=1}^{4} \text{MSE}(M_i, M_i^{GT}) + \text{MSE}(M_0, M^{GT}), \quad (7)$$

where $M^{GT}$ denotes the ground-truth local blur mask, and $i$ denotes the $i$-th level in MUGNet.

**Reconstrction loss $\mathcal{L}_{rec}$.** $\mathcal{L}_{rec}$ is proposed to regularize the deblurred results at the pixel level via

$$\mathcal{L}_{rec} = \sum_{i=1}^{4} \text{MSE}(\hat{I}_i, I_i^{GT}) + \text{MSE}(\hat{I}, I^{GT}), \quad (8)$$

where $\hat{I}_i = M_i B_i + B_i$ denotes the multi-scale latent clear result at the $i$-th level.

**SSIM loss $\mathcal{L}_{ssim}$.** The SSIM loss is also proposed to regularize the deblurred results at the pixel level via

$$\mathcal{L}_{ssim} = (1 - \sum_{i=1}^{4} \text{SSIM}(\hat{I}_i, I_i^{GT})) + (1 - \text{SSIM}(\hat{I}, I^{GT})), \quad (9)$$

where $\text{SSIM}(\cdot, \cdot)$ denotes the SSIM value.

**Multi-scale frequency reconstruction (MSFR) loss $\mathcal{L}_{mf}$.** The MSFR loss measures the $L_1$ distance between multi-scale ground-truth and deblurred images in the frequency domain as follows

$$\mathcal{L}_{mf} = \sum_{i=1}^{4} \frac{1}{4} \text{MSE}(f(\hat{I}_i), f(I_i^{GT})), \quad (10)$$

where $f(\cdot)$ denotes the fast Fourier transform that transfers the image signal to the frequency domain.

The final loss function for training our MUGNet is determined as follows

$$\mathcal{L} = \mathcal{L}_M + \lambda_1 \mathcal{L}_{rec} + \lambda_2 \mathcal{L}_{ssim} + \lambda_3 \mathcal{L}_{mf}, \quad (11)$$

where $\lambda_1$, $\lambda_2$, $\lambda_3$ are weighting factors.

## 4 Experiments

### 4.1 Experimantal Setting

**Datasets.** We follow the method in [21] and utilize the ReLoBlur dataset [21], for training and validation. We split the ReLoBlur dataset into 2,010 pairs for training and 395 pairs for testing, without repeated scenes occurring in each split set.

**Test setting.** We evaluate our MUGNet and baseline methods on ReLoBlur testing data with the full image size of $2152 \times 1436$. To quantitatively evaluate the reconstructed results, we use the Peak Signal-to-Noise Ratio (PSNR) and Structural Similarity Index (SSIM) metrics. We follow [21] and calculate the weighted PSNR, the weighted SSIM [32], and the aligned PSNR [11] specifically for the blurred regions.

**Implementation details.** We employ the Adam optimizer with parameters $\beta_1 = 0.9$ and $\beta_2 = 0.999$ to train MUGNet. The hyperparameters are set empirically as follows: $K = 50$, $\lambda_1 = 100$, $\lambda_2 = 100$, and $\lambda_3 = 10$. The batch size is configured as 12, and the initial learning rate is set to $1 \times 10^{-4}$, which is halved every 100k steps until reaching 300k steps. All training experiments are conducted using PyTorch on an NVIDIA 3090 GPU. During the data sampling process, we employ a blur-aware patch crop strategy [21]. This strategy involves sampling 50% of the training data as blur regions and the remaining 50% as random regions from the $256 \times 256$ training samples, with the assistance of blur mask annotations. It is worth noting that the model configuration for the baseline methods adheres to their respective original specifications.

### 4.2 Quantitative and Qualitative Comparisons

We conduct a comprehensive evaluation of the proposed MUGNet using the ReLoBlur dataset, benchmarking it against the following existing methods: (1) CNN-based global motion deblurring methods: DeepDeblur [27], DeblurGAN-v2 [19], SRN-DeblurNet [36], MIMO-UNet [6] and HINet [5]. (2) Transformer-based global motion deblurring methods: Restormer [52] and Uformer [41]. (3) State-of-the-art local motion deblurring method LBAG [21] and its variant LBAG+ [21] pretrained with MIMO-Unet.

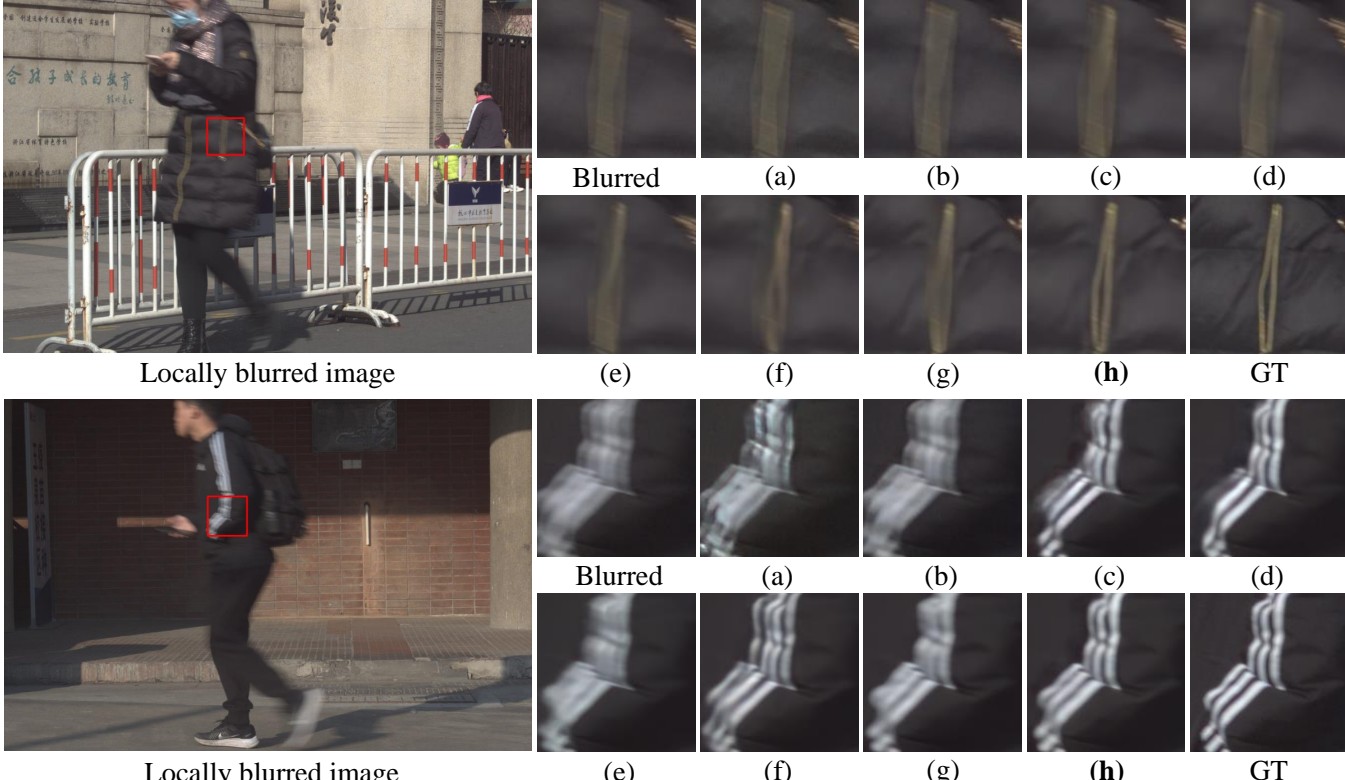

**Figure 5: Visual comparisons of state-of-the-art methods on the ReLoBlur dataset. (a) DeepDeblur [27]; (b) DeblurGAN-v2 [19]; (c) HINet [5]; (d) MIMO-UNet [6]; (e) LBAG [21]; (f) Restormer [52]; (g) Uformer [41]; (h) MUGNet (ours). Please zoom in for better visualization and best viewed on screen.**

As depicted in Table 1, MUGNet significantly excels in both PSNR and SSIM metrics. With a PSNR of 36.31 dB, it notably surpasses its nearest competitor, Uformer, by 1.12 dB. Moreover, MUGNet achieves a substantial PSNR improvement of 1.46 dB over the leading state-of-the-art local motion deblurring method, LBAG. In terms of SSIM, MUGNet outperforms Transformer-based methods by 0.0058 and surpasses LBAG+ by 0.0066, underlining its remarkable prowess in local motion deblurring. Similarly, in terms of both $PSNR_w$ and $PSNR_a$ metrics, MUGNet achieves the most superior results. Despite MUGNet exhibiting a slight reduction of 0.0013 in $SSIM_w$ compared to Uformer, it is noteworthy that our model employs only approximately half of the parameters and incurs a lower computational load. This efficient utilization of resources underscores the practical viability and computational advantage of MUGNet.

As demonstrated in Figure 5, MUGNet exhibits a remarkable superiority over its contemporary counterparts, yielding images characterized by enhanced clarity and intricate details. Notably, there is a substantial reduction in blur, resulting in the faithful preservation of intricate patterns, such as the pristine white stripes on the suit and the intricate nuances of the zipper. These restored details closely resemble the ground truth; notably, no discernible artifacts are introduced. These significant enhancements in visual

**Table 2: The ablation results of the MUQ module and the M2SA module in MUGNet.**

| ID | MUQ | M2SA | ↑PSNR | ↑SSIM | ↑$PSNR_w$ | ↑$SSIM_w$ | ↑$PSNR_a$ |
|---|---|---|---|---|---|---|---|
| (a) | ✗ | ✗ | 34.75 | 0.9267 | 28.05 | 0.8661 | 35.35 |
| (b) | ✗ | ✓ | 35.52 | 0.9294 | 28.83 | 0.8752 | 35.94 |
| (c) | ✓ | ✗ | 35.67 | 0.9299 | 29.19 | 0.8804 | 36.14 |
| (d) | ✓ | ✓ | 36.31 | 0.9323 | 30.28 | 0.8898 | 36.73 |

effects are strong evidence of MUGNet's efficiency in local motion deblurring.

### 4.3 Ablation Studies

**Effectiveness of each module.** We conduct experiments to showcase the contributions of the two core modules in our proposed MUGNet. We design the following ablations: (a) We remove both the MUQ and M2SA modules from MUGNet and replace them with residual blocks of the same parameters. (b) We remove the MUQ modules from MUGNet and replace them with residual blocks. (c) We remove M2SA modules from MUGNet and replace them with residual blocks. The ablation results are presented in Table 2, accompanied by a representative visual comparison shown in Figure 2. Comparing the results of case (d) with both components to case (a),

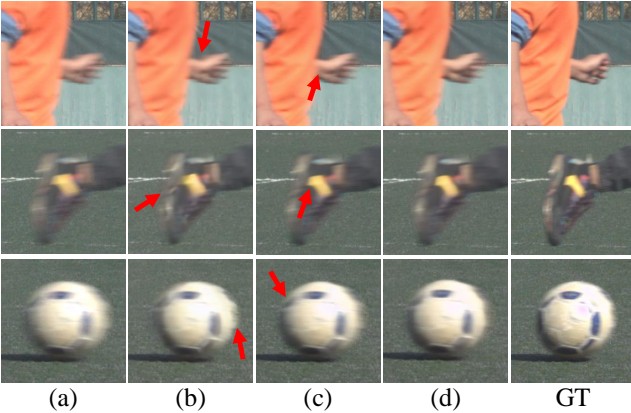

(a)  (b)  (c)  (d)  GT

**Figure 6: Analysis of the MUQ module and the proposed M2SA. (a) MUGNet w/o both components, (b) MUGNet w/o M2SA, (c) MUGNet w/o MUQ, and (d) MUGNet.**

we observe substantial improvements across all evaluation metrics. This performance gain underscores the significant contributions of both components to the overall enhancement. From the visual evidence in Figure 6, it becomes evident that removing the MUQ module results in noticeable artifacts along the edges of moving objects, while the absence of the M2SA module leads to a loss of fine details in the output. This visual observation harmonizes perfectly with our analysis and the results presented in the tables. It underscores the importance of incorporating motion uncertainty for effective local motion deblurring and further highlights the significance of the M2SA module in enhancing feature representation. This underlines the utility of the M2SA module in improving local motion deblurring results.

**A close look at the MUQ module.** To understand how the MUQ module works, we visualize the motion uncertainty maps estimated by this module in Figure 7. Our motion uncertainty maps consistently highlight object boundaries, motion boundaries, and regions with indistinguishable textures. By zooming in, one can see that these boundaries indeed mark the moving objects. As the scale increases, imprecise edges are gradually reduced, resulting in a more accurate blur mask at the largest scale. This coarse-to-fine design enhances precision in blur mask estimation. This phenomenon suggests that these particular regions tend to confuse the deblurring model. The uncertainty quantification process serves meaningful purposes: i) it enhances the interpretability of local motion deblurring models, and ii) it exposes the limitations of conventional solutions. The mask generator is trained to focus on these uncertain regions, resulting in significant improvements in accuracy when estimating clear and precise blur masks, as demonstrated in Figure 7. Additionally, we have conducted experiments to assess the influence of varying $K$ on the final results. Results are shown in Table 3. Higher values of $K$ tend to yield improved outcomes. However, taking computational costs into account, we set $K = 50$.

**A close look at the M2SA module.** Table 4 provides an analysis of the two constituents of the M2SA module, namely MTA and GTA. Within the proposed M2SA module, we employ motion-masked separable attention to enhance the feature representations

**Table 3: Ablation results of the number of samples in the MUQ.**

| $K$ | ↑PSNR | ↑SSIM | ↑PSNR$_w$ | ↑SSIM$_w$ | ↑PSNR$_a$ |
|---|---|---|---|---|---|
| 10 | 35.96 | 0.9313 | 29.53 | 0.8840 | 36.38 |
| 25 | 36.14 | 0.9311 | 29.82 | 0.8842 | 36.61 |
| 50 | 36.31 | 0.9323 | 30.28 | 0.8898 | 36.73 |
| 100 | 36.33 | 0.9326 | 30.26 | 0.8901 | 36.63 |
| 200 | 36.39 | 0.9328 | 30.29 | 0.8881 | 36.77 |

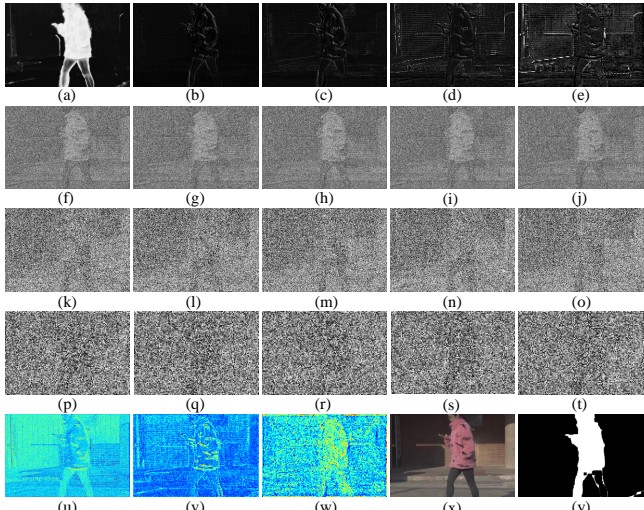

**Figure 7: (a) $M_0$, (b) $M_1$, (c) $M_2$, (d) $M_3$, (e) $M_4$, (f)-(j) five estimates sampled from $M_{init}^2$, (k)-(o) five estimates sampled from $M_{init}^3$, (p)-(t) five estimates sampled from $M_{init}^4$, (u) $U_2$, (v) $U_3$, (w) $U_4$, (x) $\hat{I}$ and (y) ground-truth blur mask.**

**Table 4: The ablation results of the proposed M2SA.**

| ID | MTA | GTA | ↑PSNR | ↑SSIM | ↑PSNR$_w$ | ↑SSIM$_w$ | ↑PSNR$_a$ |
|---|---|---|---|---|---|---|---|
| (a) | ✗ | ✗ | 35.67 | 0.9299 | 29.19 | 0.8804 | 36.14 |
| (b) | ✓ | ✗ | 36.20 | 0.9323 | 30.10 | 0.8884 | 36.61 |
| (c) | ✗ | ✓ | 36.23 | 0.9315 | 29.96 | 0.8868 | 36.63 |
| (d) | ✓ | ✓ | 36.31 | 0.9323 | 30.28 | 0.8898 | 36.70 |

within the regions affected by local motion blur. In contrast, global attention is employed to refine feature representations in static background areas. This segregation allows MUGNet to more effectively discern and address intricate patterns and specific details unique to each region. This refined modeling of local and global contexts significantly contributes to clear deblurred results.

Our M2SA module is based on Restormer. We replace the M2SA with a Transformer block from SwinIR, and the results are presented in Table 5. It can be observed that our M2SA outperforms the SwinIR Transformer block.

**Table 5: The ablation results of the M2SA and the Swin Transformer.**

| Methods | ↑PSNR | ↑SSIM | ↑$PSNR_w$ | ↑$SSIM_w$ | ↑$PSNR_a$ |
|---|---|---|---|---|---|
| Swin-Former | 36.24 | 0.9317 | 29.99 | 0.8870 | 36.65 |
| M2SA | 36.31 | 0.9323 | 30.28 | 0.8898 | 36.70 |

## 4.4 Extending MUGNet to Global Motion Deblurring

Global motion deblurring is a critical task, and despite MUGNet's initial design for local motion deblurring, it proves to be effective in global motion deblurring. We retrain MUGNet and compare its performance with advanced methods. Table 6 presents quantitative evaluations on the GoPro dataset, comparing various deblurring methods in terms of PSNR, SSIM, and the number of parameters. Notably, our proposed method, labeled as "Ours" in the table, achieves competitive results compared to state-of-the-art methods. Despite performing slightly lower than FFTformer in terms of PSNR and SSIM, our method still demonstrates promising performance with a PSNR of 34.01 and SSIM of 0.9628. Additionally, our method maintains a reasonable number of parameters (24.7 million), indicating its efficiency in terms of computational complexity. Overall, these results suggest that our approach holds considerable potential for addressing deblurring tasks on the GoPro dataset, showcasing its effectiveness alongside existing state-of-the-art methods.

## 4.5 Limitations and Discussions

While our proposed MUGNet demonstrates promising performance across various experiments, it does exhibit certain limitations in challenging scenarios. For instance, when dealing with objects in low light conditions, MUGNet might struggle to produce satisfactory results. As depicted in Figure 8 (left), under normal lighting conditions, our method yields satisfactory results. However, it might not perform as well under weaker lighting conditions, failing to generate clear results in darker areas. These limitations highlight the need for further exploration and refinement, especially in challenging scenarios such as low-light conditions. Another challenging scenario is the presence of extensive motion within the scene. Due to the inherent limitations of CNNs, capturing larger receptive fields and more global information, as achievable by Transformers, becomes challenging. Consequently, reconstructing scenes such as those depicted in Figure 8 (right) proves to be difficult. A feasible solution lies in exploring the temporal information provided by consecutive frames for local motion deblurring. This approach could enhance the network's ability to handle scenes with substantial motion and improve the overall deblurring performance. Like other local motion deblurring methods, our proposed MUGNet currently does not offer real-time local motion deblurring capabilities. Nonetheless, we are committed to refining our architecture to achieve real-time performance, which will enable its application in time-sensitive scenarios.

**Table 6: Quantitative evaluations on the GoPro dataset.**

| Methods | PSNR | SSIM | #Params (M) |
|---|---|---|---|
| DeblurGAN-v2 [19] | 29.55 | 0.9340 | 60.9 |
| SRN [37] | 30.26 | 0.9342 | 6.8 |
| DMPHN [55] | 31.20 | 0.9453 | 21.7 |
| SAPHN [35] | 31.85 | 0.9480 | 23.0 |
| MIMO-Unet+ [7] | 32.45 | 0.9567 | 16.1 |
| MPRNet [54] | 32.66 | 0.9589 | 20.1 |
| DeepRFT+ [26] | 33.23 | 0.9632 | 23.0 |
| Restormer [53] | 32.92 | 0.9611 | 26.1 |
| Uformer-B [42] | 33.06 | 0.9670 | 50.9 |
| Stripformer [38] | 33.08 | 0.9624 | 19.7 |
| NAFNet [4] | 33.71 | 0.9668 | 67.9 |
| FFTformer [16] | 34.21 | 0.9692 | 16.6 |
| Ours | 34.01 | 0.9628 | 24.7 |

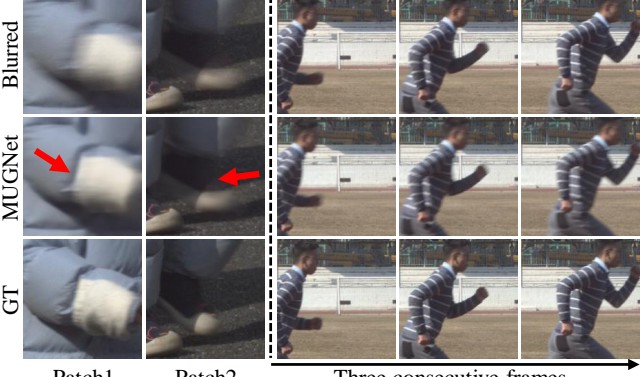

**Figure 8: Failure cases. Left: reconstruction results under normal lighting conditions and low-light scenarios. Right: the consecutive results across three frames.**

## 5 Conclusion

In this paper, we propose MUGNet to address the intricate challenges posed by motion uncertainties in the context of local motion deblurring. This is achieved through the establishment of a probabilistic representational model explicitly designed to compute these uncertainties and provide guidance to facilitate the subsequent processes. Specifically, our MUGNet comprises two key components: the MUQ module and the M2SA module. MUQ facilitates the learning of a conditional distribution that enhances the accuracy and reliability of blur map estimation, while M2SA focuses on improving the representation of regions impacted by local motion blur and the static background. We conduct extensive experiments to demonstrate the superior performance of our MUGNet.

## Acknowledgments

This project is supported by the National Research Foundation, Singapore, under its Medium Sized Center for Advanced Robotics Technology Innovation, and the Singapore Ministry of Education Academic Research Fund Tier 1 (WBS: A-0009440-01-00).

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
