# OpenReview forum: "Unraveling Motion Uncertainty for Local Motion Deblurring"
_acmmm.org/ACMMM/2024/Conference — MM2024 Poster_

### Official Review · Reviewer_AghB · 2024-05-21

**Rating:** 2
**Confidence:** 3

**Summary:**

The paper presents a novel method, the Motion Uncertainty Guided Network (MUGNet), to address the challenge of local motion deblurring. The proposed method aims to improve the accuracy and reliability of blur map estimation by incorporating a probabilistic representational model.

**Strengths:**

1. The proposed MUGNet includes MUQ module and M2SA module to solve the motion uncertainty problem in local motion deblurring.
2. MUGNet has fewer network parameters with efficient resource utilization and computational advantages.
3. The author provided performance comparisons of local motion deblurring on the ReLoBlur dataset and global motion deblurring on the GoPro dataset. They also highlighted the limitations of the proposed method, which is commendable.

**Limitations:**

(1) The main innovation of this paper lies in the design of the network framework, which introduces the MUQ module and M2SA module. However, the level of innovation requires further exploration:
1. The MUQ module claims to achieve accurate blur mask estimation for local motion deblurring by utilizing a probabilistic model to quantify motion uncertainty. While this approach is more conventional, it would benefit from a more detailed explanation of its innovative aspects.
2. The author presents M2SA as a significant contribution, but in reality, its two major components, MTA and GTA, are only minor improvements based on the Restormer's attention modules. MTA performs element-wise multiplication of the Blur map with Q and K, while GTA simply renames the components.
3. The network architecture designed by the author appears to be redundant and complex, despite having fewer parameters.

(2) Although the experimental settings follow those of LBAG, I have the following concerns:
1. In Table 1, while comparing various methods for both "local motion deblurring" and "global motion deblurring," why didn't the authors include FFTformer for comparison?
2. The analysis of the ablation experiments is comprehensive, which deserves praise. However, regarding Table 4(c) and (d), the MTA module, which is an important component of M2SA (receiving the Blur map), shows minimal improvement in performance. This aspect should be analyzed in more detail.

**Suitability:**

2

---

### Official Review · Reviewer_5Kb1 · 2024-05-24

**Rating:** 5
**Confidence:** 4

**Summary:**

This paper aims to address the local motion deblurring problems. The authors propose MUGNet including MUQ and M2SA as key modules. MUQ learns a conditional distribution of motion blur maps. M2SA separates the attention heads into MTA and GTA to enhance the representation of blurry areas and promote global interactions. Experiments on local and global blurry images are conducted.

**Strengths:**

(1) Two novel local motion deblurring modules are introduced: MUQ generates motion blur uncertainty maps, and M2SA improves blurry areas and static background regions by calculating their attention scores separately using the estimated blur masks.

(2) Experiments show promising local motion deblurring performances.

**Limitations:**

(1) What mask is used to supervise the learning of motion blur uncertainty maps? Please cite.

(2) The attention masks do not save computation.

(3)  Since the attention operation is conducted on images with 256*256 resolution while training. During inference, is attention done on the entire image?

(4) Many typos such as "based the Multi-Dconv Head Transposed Attention (TA)" in line 134, and "M2SA module" in line 922.

(5) Many expressions have been seen in published papers like Uformer, LBAG, and LMD-ViT. For example: "local motion blur is constrained to specific regions within the image" (line 48); "a straightforward strategy is to apply global deblurring techniques to local blurred images" (lines 87-89). It would be better if this paper could express a more original understanding of local motion blur. And the introduction part could be more concise.

**Suitability:**

3

---

### Official Review · Reviewer_XpAp · 2024-06-09

**Rating:** 4
**Confidence:** 3

**Summary:**

In the submitted manuscript, the authors propose to tackle the local motion deblurring task by leveraging a probabilistic representational model MUGNet. The Motion-Uncertainty Quantification (MUQ) module and Motion-Masked Separable Attention (M2SA) module are further designed to estimate the accurate conditional distribution and enhance the regions affected by local motion blur, respectively. The extensive numerical and visualized results are presented to demonstrate the effectiveness of the method. The paper exhibits logical coherence and the accompanying illustrations facilitate comprehension; however, further clarification is required regarding the innovativeness of the module design.

**Strengths:**

1. The paper is well-organized and easy to follow.

2. The proposed method is technically feasible and provides a comprehensive review of the relevant work.

3. The sufficient quantitative and qualitative evaluations are provided to substantiate the efficacy of the method.

4. The paper provides a detailed analysis of the limitations and presents several failure cases for future directions.

**Limitations:**

1. My main concern is the innovation of the Motion-Masked Separable Attention (M2SA) Module. Combining the global spatial information and masked foreground information by the self-attention based approach seems to be common in the previous computer vision tasks. Perhaps further specific improvements are needed.

2. Additionally, the structural design of the Motion Uncertainty Quantification Module is widely utilized in Variational AutoEncoder (VAE) or its variants. Further analysis is needed to determine the improvements made in this regard.

3. The MUQ module is designed to estimate the motion uncertainty map. Have any alternative uncertainty estimation methods been attempted for the sake of comparative analysis?

4. Other minor mistakes or improvements, e.g., 1) There is an extra "and" in the caption "... the best and and the second ..." of Table 1;  2) The SSIMw of  LMD-ViT method ("0.8938") should be in bold in Table 1, not the value of Uformer-B ("0.8911"); 3) The last line of PSNRa in Table 4 is mismatched with the previous results, which should be "36.73";  4) The Swin-Former in Table 5 lacks reference.

**Suitability:**

2

---

### Official Review · Reviewer_kQWF · 2024-06-09

**Rating:** 4
**Confidence:** 3

**Summary:**

The paper proposes a Motion-Uncertainty-Guided Network (MUGNet) that addresses motion uncertainties with a probabilistic model, comprising motion-uncertainty quantification and motion-masked separable attention modules. MUGNet outperforms existing solutions, as demonstrated through extensive experiments.

**Strengths:**

1. This paper is easy-to-follow, with detailed introduction for the method and evaluation.
2. The proposed MUQ module works as a probabilistic model for motion uncertainty quantification, which is an inspiring idea.

**Limitations:**

1. All quantitative experimental results lack the inclusion of LPIPS, a widely-utilized metric for perceptually evaluating restoration outcomes in low-level methods. The absence of LPIPS considerably undermines the experiments' credibility.
2. Table 1 contains a mis-notation for SSIM_w. MUGNet does not attain the second-best results, as LMD-ViT yields higher scores of 0.8938.
3. The qualitative outcomes of LMD-ViT are omitted in Fig. 5. Furthermore, based on Fig. 3 from LMD-ViT's paper, it appears that the reported result in LMD-ViT's paper (the first case) is quite impressive and closely resembles the MUGNet result shown in Fig. 5 of this submission (the second case). A clarification for this is required in the rebuttal.

I am inclined to accept this submission if the rebuttal addresses the following concerns:
1. providing LPIPS results for quantitative experiments.
2. Issues related to the qualitative results in Fig. 5.

**Suitability:**

2

---

### Official Review · Reviewer_D5Ut · 2024-06-09

**Rating:** 4
**Confidence:** 2

**Summary:**

The paper proposes a novel network structure to deal with local motion blur in an image.
The network learns to estimate pixel-wise probability representation maps for blur masks on multiple scales, with reparameterization trick to enforce tractability for gradient calculation during training phase.
Global and local attention modules are introduced to further facilitate blur removal.
Comprehensive experiments are carried out to prove the advantage of the proposed method.

**Strengths:**

- Novelty in probability representation for blur mask estimation. A spectrum of uncertainty allows larger margin of error for each module than binarized maps. The design also incorporates Bayes' theorem for a robust process, which hasn't been widely adopted for such tasks.
- Extended experiment results to support the paper's contributions. The qualitative and quantitative results are convincing, with several ablation studies to offer more thorough view on each module.

**Limitations:**

- Accuracy of blur mask estimation is not clear. Although carried out in a probabilistic fashion, actual results of such estimations shown in Figure 7 do not have distinct alignment with intuition. Except of subplot (a) in the Figure, the rest show more noise pattern than useful information, which brings up a question of whether the network is estimation blur mask or not.
- Marginal improvements. Though demonstrating performance gain, exact numbers of metrics increase are not huge. Considering the fact that no obvious runtime is reduced, the cost-effectiveness of the model design remains questionable.

**Suitability:**

2

---

### Meta-Review · Area_Chair_1cSL · 2024-07-05

**Recommendation:** Accept (Poster)
**Confidence:** 5

**Metareview:**

Although this paper received mixed scores in the first round of review, the rebuttal has successfully reversed the opinions of the reviewer on the negative side. AC believes that this paper should be accepted, and encourages authors to improve the camera ready as they said in the rebuttal.